# The Space Complexity of Approximating Logistic Loss

**Gregory Dexter**
LinkedIn Corporation
gdexter@linkedin.com

**Petros Drineas**
Department of Computer Science
Purdue University
pdrineas@purdue.edu

**Rajiv Khanna**
Department of Computer Science
Purdue University
rajivak@purdue.edu

## Abstract

We provide space complexity lower bounds for data structures that approximate logistic loss up to $\epsilon$-relative error on a logistic regression problem with data $\mathbf{X} \in \mathbb{R}^{n \times d}$ and labels $\mathbf{y} \in \{-1, 1\}^d$. The space complexity of existing coreset constructions depend on a natural complexity measure $\mu_{\mathbf{y}}(\mathbf{X})$, first defined in [10]. We give an $\tilde{\Omega}(\frac{d}{\epsilon^2})$ space complexity lower bound in the regime $\mu_{\mathbf{y}}(\mathbf{X}) = \mathcal{O}(1)$ that shows existing coresets are optimal in this regime up to lower order factors. We also prove a general $\tilde{\Omega}(d \cdot \mu_{\mathbf{y}}(\mathbf{X}))$ space lower bound when $\epsilon$ is constant, showing that the dependency on $\mu_{\mathbf{y}}(\mathbf{X})$ is not an artifact of mergeable coresets. Finally, we refute a prior conjecture that $\mu_{\mathbf{y}}(\mathbf{X})$ is hard to compute by providing an efficient linear programming formulation, and we empirically compare our algorithm to prior approximate methods.

## 1   Introduction

Logistic regression is an indispensable tool in data analysis, dating back to at least the early 19th century. It was originally used to make predictions in social science and chemistry applications [14, 15, 3], but over the past 200 years it has been applied to all data-driven scientific domains, from economics and the social sciences to physics, medicine, and biology. At a high level, the (binary) logistic regression model is a statistical abstraction that models the probability of one of two alternatives or classes by expressing the log-odds (the logarithm of the odds) for the class as a linear combination of one or more predictor variables.

Formally, logistic regression aims to find a parameter vector $\beta \in \mathbb{R}^d$ that minimizes the logistic loss, $\mathcal{L}(\beta)$, which is defined as follows:

$$\mathcal{L}(\beta) = \sum_{i=1}^{n} \log(1 + e^{-\mathbf{y}_i \mathbf{x}_i^T \beta}), \tag{1}$$

where $\mathbf{X} \in \mathbb{R}^{n \times d}$ is the data matrix ($n$ points in $\mathbb{R}^d$, with $\mathbf{x}_i^T$ being the rows of $\mathbf{X}$) and $\mathbf{y} \in \{-1, 1\}^n$ is the vector of labels whose entries are the $\mathbf{y}_i$. Due to the central importance of logistic regression, algorithms and methods to improve the efficiency of minimizing the logistic loss are always of interest [7, 10, 9].

The prior study of linear regression, a much simpler problem that admits a closed-form solution, has provided ample guidance on how we may expect to improve the efficiency of logistic regression. Let

38th Conference on Neural Information Processing Systems (NeurIPS 2024).

us first consider how a data structure that approximates $\ell_2$-regression loss may be leveraged to design efficient algorithms for linear regression. Let $\mathcal{D}(\cdot) : \mathbb{R}^d \to \mathbb{R}$ be a data structure such that:

$$(1 - \epsilon)\|\mathbf{A}\mathbf{x} - \mathbf{b}\|_2 \le \mathcal{D}(\mathbf{x}) \le (1 + \epsilon)\|\mathbf{A}\mathbf{x} - \mathbf{b}\|_2.$$

Then, for, say, any $\epsilon \in (0, 1/3)$

$$\tilde{\mathbf{x}} = \underset{\mathbf{x} \in \mathbb{R}^d}{\operatorname{argmin}} \mathcal{D}(\mathbf{x}) \Rightarrow \|\mathbf{A}\tilde{\mathbf{x}} - \mathbf{b}\|_2 \le \frac{1 + \epsilon}{1 - \epsilon} \cdot \min_{\mathbf{x} \in \mathbb{R}^d} \|\mathbf{A}\mathbf{x} - \mathbf{b}\|_2$$
$$\le (1 + 3\epsilon) \cdot \min_{\mathbf{x} \in \mathbb{R}^d} \|\mathbf{A}\mathbf{x} - \mathbf{b}\|_2.$$

That is, a data structure that approximates the $\ell_2$-regression loss up to $\epsilon$-relative error may be used to solve the original regression problem up to $3\epsilon$-relative error. This is particularly interesting when $\mathcal{D}(\cdot)$ has lower *space* complexity than the original problem or can be minimized more efficiently.

Practically efficient data structures satisfying these criteria for linear regression have been instantiated through matrix sketching and leverage score sampling [16]. There is extensive work exploring constructions of a matrix $\mathbf{S} \in \mathbb{R}^{s \times n}$, where given a data matrix $\mathbf{A} \in \mathbb{R}^{n \times d}$ and vector of labels $\mathbf{b} \in \mathbb{R}^n$, we may solve the lower dimensional problem $\tilde{\mathbf{x}} = \operatorname{argmin}_{\mathbf{x} \in \mathbb{R}^d} \|\mathbf{S}\mathbf{A}\mathbf{x} - \mathbf{S}\mathbf{b}\|_2$ to achieve the guarantee $\|\mathbf{A}\tilde{\mathbf{x}} - \mathbf{b}\|_2 \le \frac{1 + \epsilon}{1 - \epsilon} \cdot \min_{\mathbf{x} \in \mathbb{R}^d} \|\mathbf{A}\mathbf{x} - \mathbf{b}\|_2$ for a chosen $\epsilon > 0$. Under conditions that guarantee that $s \ll n$, we can achieve significant *computational time and space savings* by following such an approach. An important class of matrices $\mathbf{S} \in \mathbb{R}^{s \times n}$ that guarantee the above approximation are the so-called $\ell_2$-subspace embeddings which satisfy:

$$(1 - \epsilon)\|\mathbf{A}\mathbf{x} - \mathbf{b}\|_2 \le \|\mathbf{S}(\mathbf{A}\mathbf{x} - \mathbf{b})\|_2 \le (1 + \epsilon)\|\mathbf{A}\mathbf{x} - \mathbf{b}\|_2 \quad \text{for all } \mathbf{x} \in \mathbb{R}^d.$$

Despite the central importance of logistic regression in statistics and machine learning, relatively little is known about how the method behaves when matrix sketching and sampling are applied to its input. Munteanu et al. [10, 11] initiated the study of *coresets* for logistic regression. Meanwhile, Munteanu and Omlor [9] provide the current state-of-the-art bounds bounds on the size of a coreset for logistic regression. Analogously to linear regression, these works present an efficient data structure $\tilde{\mathcal{L}}(\cdot)$ such that

$$(1 - \epsilon)\mathcal{L}(\beta) \le \tilde{\mathcal{L}}(\beta) \le (1 + \epsilon)\mathcal{L}(\beta). \tag{2}$$

We call $\tilde{\mathcal{L}}(\cdot)$ an $\epsilon$-*relative error approximation* to the logistic loss. Prior work on coreset construction for logistic regression critically depends on the data complexity measure $\mu_{\mathbf{y}}(\mathbf{X})$, which was first introduced in [10], and is defined as follows.

**Definition 1.** *(Classification Complexity Measure [10]) For any $\mathbf{X} \in \mathbb{R}^{n \times d}$ and $\mathbf{y} \in \{-1, 1\}^n$, let*

$$\mu_{\mathbf{y}}(\mathbf{X}) = \sup_{\beta \ne \mathbf{0}} \frac{\|(\mathbf{D}_{\mathbf{y}}\mathbf{X}\beta)^+\|_1}{\|(\mathbf{D}_{\mathbf{y}}\mathbf{X}\beta)^-\|_1},$$

*where $\mathbf{D}_{\mathbf{y}}$ is a diagonal matrix with $\mathbf{y}$ as its diagonal, and $(\mathbf{D}_{\mathbf{y}}\mathbf{X}\beta)^+$ and $(\mathbf{D}_{\mathbf{y}}\mathbf{X}\beta)^-$ denote the positive and the negative entries of $\mathbf{D}_{\mathbf{y}}\mathbf{X}\beta$ respectively.*

Specifically, these methods construct a coreset by storing a subset of the rows indexed by $\mathcal{S} \subset \{1 \ldots n\}$ such that $|\mathcal{S}| = \tilde{\mathcal{O}}(\frac{d \cdot \mu_{\mathbf{y}}(\mathbf{X})}{\epsilon^2})$ and generating a set of weights $\{w_i\}_{i \in \mathcal{S}}$ such that each $w_i$ is specified by $\mathcal{O}(\log nd)$ bits [9]. The approximate logistic loss is then computed as:

$$\tilde{\mathcal{L}}(\beta) = \sum_{i \in \mathcal{S}} w_i \cdot \log(1 + e^{-\mathbf{y}_i \mathbf{x}_i^T \beta}) \tag{3}$$

is an $\epsilon$-relative error approximation to $\mathcal{L}(\beta)$. We see that $\mu_{\mathbf{y}}(\mathbf{X})$ is important in determining how compressible a logistic regression problem is through coresets, and prior has proven this dependency in coresets is necessary [17]. Our work further shows this dependency is fundamental to the space complexity of approximating logistic loss by any data structure.

Our work advances understanding of data structures that approximate logistic loss to reduce its space and time complexity. Our results provide guidance on how existing coreset constructions may be improved upon as well as their fundamental limitations.

## 1.1 Our contributions

We briefly summarize our contributions in this work; see Section 1.3 for notation.

- We prove that any data structure that approximates logistic loss up to $\epsilon$-relative error must use $\tilde{\Omega}(\frac{d}{\epsilon^2})$ space in the worst case on a dataset with constant $\mu$-complexity (Theorem 1).

- We show that any coreset that provides an $\epsilon$-relative error approximation to logistic loss requires storing $\tilde{\Omega}(\frac{d}{\epsilon^2})$ rows of the original data matrix $\mathbf{X}$ (Corollary 2). Thereby, we prove that analyses of existing coreset constructions are optimal up to logarithmic factors in the regime where $\mu_{\mathbf{y}}(\mathbf{X}) = \mathcal{O}(1)$.

- We prove that *any* data structure that approximates logistic loss to relative error must take $\tilde{\Omega}(d \cdot \mu_{\mathbf{y}}(\mathbf{X}))$ space, thereby showing that the dependency on the $\mu$-complexity measure is not an artifact of using mergeable coresets which the prior work [17] had relied on (Theorem 3).

- We provide experiments that demonstrate how prior methods that only approximate $\mu_{\mathbf{y}}(\mathbf{X})$ can be substantially inaccurate, despite being more complicated to implement than our method (see Section 4).

- Finally, we prove that low rank approximations can provide a simple but weak additive error approximation to logistic loss and these guarantees are tight in the worst case (See Appendix D).

## 1.2 Related Work

Prior work has explored the space complexity of data structures that preserve $\|\mathbf{X}\beta\|_p$ for $\beta \in \mathbb{R}^d$, particularly in the important case where $p = 2$. Lower bounds for this problem are analogous to our work and motivate our inquiry. An early example of such work is [12], which lower bounds the minimum dimension of an "oblivious subspace embedding", a particular type of data structure construction that preserves $\|\mathbf{X}\beta\|_2$. A more recent example in this line of work is [6], which provides space complexity lower bounds for general data structures that preserves $\|\mathbf{X}\beta\|_p$ for general $p \in \mathbb{N}$.

Recent work on the development of coresets for logistic regression motivates our focus on this problem. This line of work was initiated by Munteanu et al. [10]. Mai et al. [7] used Lewis weight sampling to achieve an $\tilde{\mathcal{O}}(d\mu_{\mathbf{y}}(\mathbf{X})^2 \cdot \epsilon^{-2})$. The work of Woodruff and Yasuda [17] later provided an online coreset construction containing $\tilde{\mathcal{O}}(d\mu_{\mathbf{y}}(\mathbf{X})^2 \cdot \epsilon^{-2})$ points as well as a coreset construction using $\tilde{\mathcal{O}}(d^2\mu_{\mathbf{y}}(\mathbf{X}) \cdot \epsilon^{-2})$ points. Finally, Munteanu and Omlor [9] recently proved an $\tilde{\mathcal{O}}(d\mu_{\mathbf{y}}(\mathbf{X}) \cdot \epsilon^{-2})$ size coreset construction. Our work is complementary to the above works, since it highlights the limits of possible compression of the logistic regression problem while maintaining approximation guarantees to the original problem.

## 1.3 Notation

We assume, without loss of generality, that $\mathbf{y}_i = -1$ for all $i = 1 \ldots n$. Any logistic regression problem with $(\mathbf{X}, \mathbf{y})$ defined above can be transformed to this standard form by multiplying both $\mathbf{X}$ and $\mathbf{y}$ by the matrix $-\mathbf{D_y}$. Here $\mathbf{D_y} \in \mathbb{R}^{n \times n}$ is a diagonal matrix with $i$-th entry set as $\mathbf{y}_i$. The logistic loss of the original problem is equal to that of the transformed problem for all $\beta \in \mathbb{R}^d$. Let $\mathbf{M}_i$ denote the $i$-th row of a matrix $\mathbf{M}$. We denote as $\mathbf{X}$ the matrix formed by stacking $\mathbf{x}_i$ as rows. We will use $\tilde{\mathcal{O}}(\cdot)$ and $\tilde{\Omega}(\cdot)$ to suppress logarithmic factors of $d$, $n$, $1/\epsilon$, and $\mu_{\mathbf{y}}(\mathbf{X})$. Finally, let $[n] = \{1, 2, ..., n\}$.

## 2 Space complexity lower bounds

In this section, we provide space complexity lower bounds for a data structure $\tilde{\mathcal{L}}(\cdot)$ that satisfies the relative error guarantee in eqn. 2. We use the notations as specified in Section 1. Additionally, we require throughout this section that the entries of $\mathbf{X}$ can be specified in $\mathcal{O}(\log nd)$ bits.

Our first main result is a general lower bound on the space complexity of *any* data structure which approximates logistic loss to $\epsilon$-relative error for every parameter vector $\beta \in \mathbb{R}^d$ on a data set whose

complexity measure $\mu_{\mathbf{y}}(\mathbf{X})$ is bounded by a constant. As a corollary to this result, we show that existing coreset constructions are optimal up to lower order terms in the regime where $\mu_{\mathbf{y}}(\mathbf{X}) = \mathcal{O}(1)$. Our second main result shows that any data structure providing a $\epsilon_0$-constant factor approximation to the logistic loss requires $\tilde{\Omega}(\mu_{\mathbf{y}}(\mathbf{X}))$ space, where $\epsilon_0 > 0$ is constant. Both of these results proceed by reduction to the INDEX problem [6, 8] (see Section A.2), where we use the fact that an approximation to the logistic loss can approximate ReLU loss under appropriate conditions.

Consider the ReLU loss:

$$\mathcal{R}(\beta; \mathbf{X}) = \sum_{i=1}^{n} \max\{\mathbf{x}_i^T \beta, 0\}. \tag{4}$$

Our lower bound reductions hinge on the fact that a relative error approximation to logistic loss can be used to simulate a relative error approximation to ReLU loss under appropriate conditions. We capture this in the following lemma. We include all proofs omitted from the main text in Appendix B.

**Lemma 2.1.** *Given a data set $\mathbf{X} \in \mathbb{R}^{n \times d}$ and $\mathcal{B} \subset \mathbb{R}^d$ such that $\inf_{\beta \in \mathcal{B}} \mathcal{R}(\beta; \mathbf{X}) > 1$, if there exists a data structure $\tilde{\mathcal{L}}(\cdot)$ that satisfies:*

$$(1 - \epsilon)\mathcal{L}(\beta) \leq \tilde{\mathcal{L}}(\beta) \leq (1 + \epsilon)\mathcal{L}(\beta) \text{ for all } \beta \in \mathcal{B},$$

*then there exists a data structure taking $\tilde{\mathcal{O}}(1)$ extra space such that:*

$$(1 - 3\epsilon)\mathcal{R}(\beta) \leq \tilde{\mathcal{R}}(\beta) \leq (1 + 3\epsilon)\mathcal{R}(\beta) \text{ for all } \beta \in \mathcal{B}.$$

## 2.1 Lower bounds for the $\mu_{\mathbf{y}}(\mathbf{X}) = \Theta(1)$ regime

We lower bound the space complexity of any data structure that approximates logistic loss to $\epsilon$-relative error. Recall that the running time of the computation compressing the data to a small number of bits and evaluating $\tilde{\mathcal{L}}(\beta)$ for a given query $\beta$ is unbounded in this model. Hence, Theorem 1 provides a strong lower bound on the space needed for *any* compression of the data that can be used to compute an $\epsilon$-relative error approximation to the logistic loss, including, but not limited to, coresets.

At a high level, our proof operates by showing that a relative error approximation to logistic loss can be used to obtain a relative error approximation to ReLu regression, which in turn can be used to construct a relative error $\ell_1$-subspace embedding. Previously, Li et al. [6] lower bounded the worst case space complexity of any data structure that maintains an $\ell_1$-subspace embedding by reducing the problem to the well-known INDEX problem in communication complexity. Notably, the construction of $\mathbf{X}$ has complexity measure bounded by a constant, i.e., $\mu_{\mathbf{y}}(\mathbf{X}) = \mathcal{O}(1)$.

**Lemma 2.2.** *There exists a matrix $\mathbf{X} \in \mathbb{R}^{n \times d}$ such that $\mu_{\mathbf{y}}(\mathbf{X}) \leq 4$ and any data structure that, with at least $2/3$ probability, approximates $\mathcal{R}(\beta; \mathbf{X})$ up to $\epsilon$-relative error requires $\tilde{\Omega}(\frac{d}{\epsilon^2})$ space, provided that $d = \Omega(\log 1/\epsilon)$, $n = \tilde{\Omega}(d/\epsilon^2)$, and $0 < \epsilon \leq \epsilon_0$ for some small constant $\epsilon_0$.*

*Furthermore, $\mathcal{R}(\beta; \mathbf{X}) > 3\|\beta\|_1$ for all $\beta \in \mathbb{R}^d$.*

Using Lemma 2.1, we can extend this lower bound on the space complexity to approximate ReLU loss to logistic loss.

**Theorem 1.** *Any data structure $\tilde{\mathcal{L}}(\cdot)$ that, with at least $2/3$ probability, is an $\epsilon$-relative error approximation to logistic loss for some input $(\mathbf{X}, \mathbf{y})$, requires $\tilde{\Omega}\left(\frac{d}{\epsilon^2}\right)$ space in the worst case, assuming that $d = \Omega(\log 1/\epsilon)$, $n = \tilde{\Omega}\left(d\epsilon^{-2}\right)$, and $0 < \epsilon \leq \epsilon_0$ for some sufficiently small constant $\epsilon_0$.*

*Proof.* By Lemma 2.2, there exists a matrix $\mathbf{X}$ such that any data structure that approximates the ReLU loss up to $\epsilon$-relative error requires the stated space complexity. Let

$$\mathcal{B} = \{\beta \in \mathbb{R}^d \mid \|\beta\|_1 = 1\}$$

Then, by Lemma 2.2, $\inf_{\beta \in \mathcal{B}} \mathcal{R}(\beta; \mathbf{X}) \geq 3$. Therefore, by Lemma 2.1, any $(1 + \epsilon)$ factor approximation to the logistic loss for the matrix $\mathbf{X}$ provides a $(1 + 3\epsilon)$-factor approximation to $\mathcal{R}(\beta; \mathbf{X})$ for $\beta \in \mathcal{B}$. Since $\mathcal{R}(\beta) = \|\beta\|_1 \cdot \mathcal{R}(\beta/\|\beta\|_1)$, we can extend this guarantee to all $\beta \in \mathbb{R}^d$. By Lemma 2.2, any data structure that provides such a guarantee requires the stated space complexity, and, finally, $\tilde{\mathcal{L}}(\cdot)$ requires the stated complexity. $\square$

From the above theorem, we can derive a lower bound on the number of rows needed by a coreset that achieves an $\epsilon$-relative error approximation to the logistic loss (see eqn. 3 for specification of a coreset).

**Corollary 2.** *Any coreset construction that, with at least $2/3$ probability, satisfies the relative error guarantee in eqn. 2 must store $\tilde{\Omega}(\frac{d}{\epsilon^2})$ rows of some input matrix $\mathbf{X}$, where $\mu_{\mathbf{y}}(\mathbf{X}) = \mathcal{O}(1)$.*

*Proof.* Using the previous theorem, there exists a matrix $\mathbf{X} \in \mathbb{R}^{n \times \mathcal{O}(\log n)}$ such that, assuming that $n = \tilde{\Omega}(\epsilon^{-2})$ and $\epsilon$ is sufficiently small, any data structure that approximates the logistic loss up to relative error on $\mathbf{X}$ must use $\tilde{\Omega}(1/\epsilon^2)$ bits in the worst case. (Recall that $\tilde{\Omega}(\cdot)$ suppresses $\log n$ factors.)

If the data structure stores entire rows of $\mathbf{X}$ while storing a total of $\tilde{\Omega}(\frac{1}{\epsilon^2})$ bits, then it must store at least

$$\tilde{\Omega}\left(\frac{1}{\epsilon^2} \cdot \frac{1}{\log n}\right) = \tilde{\Omega}\left(\frac{1}{\epsilon^2}\right)$$

rows of $\mathbf{X}$ in total.

Recall that the proof of Theorem 1 proceeds by showing that a relative error approximation to the logistic loss can be used to solve the `INDEX` problem. If we have $d$ independent instances of the matrix $\mathbf{X}$, i.e., $\mathbf{X}_{(1)}, \mathbf{X}_{(2)}, ...\mathbf{X}_{(d)}$, then we may create the matrix

$$\mathbf{Y} = \begin{bmatrix} \mathbf{X}_{(1)} & \mathbf{0} & \dots & \mathbf{0} \\ \mathbf{0} & \mathbf{X}_{(2)} & \ddots & \mathbf{0} \\ \vdots & \ddots & \ddots & \mathbf{0} \\ \mathbf{0} & \mathbf{0} & \cdot & \mathbf{X}_{(d)} \end{bmatrix}.$$

Note that any relative error approximation to the logistic loss on $\mathbf{Y}$ would allow relative error approximation to the logistic loss on each of the individual $\mathbf{X}_{(i)}$, $i = 1 \dots d$ matrices, thus allowing one to solve $d$ instances of the `INDEX` problem simultaneously. This implies that we can query any bit in each of the $d$ index problems which solves an `INDEX` problem of size $\tilde{\Omega}\left(d/\epsilon^2\right)$.

If the data structure is restricted to store entire rows of $\mathbf{X}_{(i)}$, then recall that we must store $\tilde{\Omega}(1/\epsilon^2)$ rows of $\mathbf{X}_{(i)}$. Therefore, we conclude that any coreset that achieves a relative error approximation to the logistic loss on $\mathbf{Y}$ with at least $2/3$ probability must store $\tilde{\Omega}(d/\epsilon^2)$ rows of $\mathbf{Y}$. $\qquad \square$

The work of Munteanu and Omlor [9] showed that sampling $\tilde{\mathcal{O}}\left(\frac{d \cdot \mu_{\mathbf{y}}(\mathbf{X})}{\epsilon^2}\right)$ rows of $\mathbf{X}$ yields an $\epsilon$-relative error coreset for logistic loss with high probability. Hence, Corollary 2 implies that the coreset construction of Mai et al. [7] is of optimal size in the regime where $\mu_{\mathbf{y}}(\mathbf{X}) = \mathcal{O}(1)$. However, Theorem 1 only guarantees that coresets are optimal up to a $d$ factor in terms of bit complexity. An interesting future direction would be closing this gap by either proving that coresets have optimal bit complexity or showing approaches, like matrix sparsification, could achieve even greater space savings.

## 2.2 An $\tilde{\Omega}(\mu_{\mathbf{y}}(\mathbf{X}) \cdot d)$ lower bound

In this section, we provide a space complexity lower bound for a data structure achieving a constant $\epsilon_0$-relative error approximation to logistic loss on an input $\mathbf{X}$ with variable $\mu_{\mathbf{y}}(\mathbf{X})$. We again assume $\mathbf{y}_i = -1$ for all $i \in [n]$ without loss of generality.

The proof depends on the existence of a matrix $\mathbf{M} \in \{-1, 1\}^{n \times \log^4 n}$ that has nearly orthogonal rows. From $\mathbf{M}$, we can construct the matrix $\mathbf{X}$ such that $\mu_{\mathbf{y}}(\mathbf{X}) = \mathcal{O}(n)$ and a 2-factor approximation to ReLU loss on $\mathbf{X}$ can solve the size $n$ `INDEX` problem. By Lemma 2.1 and the lower space complexity bound for solving the `INDEX` problem, we prove the described lower bound for approximating logistic loss.

We begin by proving the existence of the matrix $\mathbf{M}$. Recall that for any matrix $\mathbf{M}$, we use $\mathbf{M}_i$ to denote the $i$-th row of $\mathbf{M}$.

**Lemma 2.3.** *Let $n = 2^p$ for $n \in \mathbb{N}$. There exists a matrix $\mathbf{M} \in \{-1, 1\}^{n \times k}$ such that $k = \log^4 n$ and $|\langle \mathbf{M}_i, \mathbf{M}_j \rangle| \le 4 \log^2 n$ for all $i \ne j$.*

We now use the previous lemma to construct a matrix $\mathbf{X}$ such that a 2-factor approximation to ReLU loss on $\mathbf{X}$ requires $\tilde{\Omega}(d \cdot \mu_\mathbf{y}(\mathbf{X}))$ space.

**Lemma 2.4.** *Let $n = 2^p$ for $n \in \mathbb{N}$ and assume that $\log^4(n/2) > 16 \log^2 n$. Then, there exists a matrix $\mathbf{X} \in \mathbb{R}^{n \times k}$ such that $k = \mathcal{O}(\log^4 n)$ and $\mu_\mathbf{y}(\mathbf{X}) = \mathcal{O}(n)$ such that any data structure $\tilde{\mathcal{R}}(\cdot)$ that, with at least $2/3$ probability, satisfies (for a fixed $\beta \in \mathbb{R}^d$)*

$$\mathcal{R}(\beta) \le \tilde{\mathcal{R}}(\beta) \le 2\mathcal{R}(\beta)$$

*and requires at least $\Omega(n)$ bits of space.*

*Proof.* Our proof will proceed by reduction to the INDEX problem. Let $y_i \in \{0, 1\}$ for all $i = 1 \ldots n/2]$ represent an arbitrary sequence of $n/2$ bits. We will show how to encode the state of the $n$ bits in a relative error approximation to ReLU loss on some data set $\mathbf{X}$.

First, let us start with the matrix $\mathbf{M} \in \{-1, 1\}^{n/2 \times k'}$ specified in Lemma 2.3, where $k' = \log^4(n/2)$. Let $\tilde{\mathbf{M}} \in \mathbb{R}^{n/2 \times k'}$ such that $\tilde{\mathbf{M}}_{i*} = \tilde{\mathbf{M}}_{i*}$ if $y_i = 1$ and $\tilde{\mathbf{M}}_{i*} = 1/2 \cdot \mathbf{M}_{i*}$ otherwise. In words, we multiply the $i$-th row of $\mathbf{M}$ by one if $y_i = 1$ and $1/2$ if $y_i = 0$. Next, let us construct the matrix $\mathbf{X} \in \mathbb{R}^{n \times k'+1}$:

$$\mathbf{X} = \begin{bmatrix} \tilde{\mathbf{M}} & \mathbf{1} \\ -\mu \cdot \tilde{\mathbf{M}} & -\mu \cdot \mathbf{1} \end{bmatrix}, \tag{5}$$

where $\mu > 0$ will be specified later.

Suppose we want to query $y_i$. Let $\beta = [\tilde{\mathbf{M}}_i, -4 \log^2 n]^T$. We will show that $\tilde{\mathbf{M}}_j \beta < \mu \cdot 8 \log^2 n$ for all $j \ne i$ by considering three cases:

**Case 1** ($j \le n/2$; $j \ne i$)**:** In this case, $\tilde{\mathbf{M}}_j \beta = \langle \tilde{\mathbf{M}}_i, \tilde{\mathbf{M}}_j \rangle - 4 \log^2 n$. By Lemma 2.3, $\langle \tilde{\mathbf{M}}_i, \tilde{\mathbf{M}}_j \rangle \le \langle \mathbf{M}_i, \mathbf{M}_j \rangle < 4 \log^2(n/2)$, hence we can conclude this case.

**Case 2** ($j > n/2$; $j \ne 2i$)**:** Here, $\mathbf{X}_j \beta = -\mu \langle \tilde{\mathbf{M}}_i, \tilde{\mathbf{M}}_j \rangle + 4\mu \log^2 n$. Since $|\langle \tilde{\mathbf{M}}_i, \tilde{\mathbf{M}}_j \rangle| \le |\langle \mathbf{M}_i, \mathbf{M}_j \rangle| < 4 \log^2 n$, $\mathbf{X}_j \beta < \mu \cdot 8 \log^2 n$, so we conclude the case.

**Case 3** ($j = 2i$)**:** In this case, $\mathbf{X}_j \beta = -\mu \langle \tilde{\mathbf{M}}_i, \tilde{\mathbf{M}}_i \rangle + 4\mu \log^2 n$. Since $-\mu \langle \tilde{\mathbf{M}}_i, \tilde{\mathbf{M}}_i \rangle$ is negative, we conclude the case.

The above cases show that $\mathbf{X}_j \beta < \mu \cdot 8 \log^2 n$ when $j \ne i$. Therefore,

$$\mathcal{R}(\beta) \le n \cdot \mu \cdot 8 \log^2 n + \text{ReLU}(\mathbf{X}_i \beta) \quad \text{and} \quad \mathcal{R}(\beta) \ge \text{ReLU}(\mathbf{X}_i \beta).$$

We next show that the bit $y_i$ will have a large effect on $\mathcal{R}(\beta)$. If $y_i = 0$, then,

$$\mathbf{X}_i \beta = \langle \tilde{\mathbf{M}}_i, \tilde{\mathbf{M}}_i \rangle = \frac{1}{4} \|\mathbf{M}_i\|_2^2 - 4 \log^2 n < \frac{1}{4} \cdot \log^4(n/2),$$

since $\tilde{\mathbf{M}}_i \in \mathbb{R}^{\log^4(n/2)}$. On the other hand, if $y_i = 1$, then,

$$\mathbf{X}_i \beta = \|\tilde{\mathbf{M}}_i\|_2^2 - 4 \log^2 n = \log^4(n/2) - 4 \log^2 n > \frac{3}{4} \log^4(n/2),$$

where we used our assumption that $\log^4(n/2) > 16 \log^2 n$. Therefore, if $y_i = 0$, $\mathcal{R}(\beta) \le \frac{1}{4} \cdot \log^4 n + n \cdot 8\mu \log^2 n$. By setting $\mu = \frac{\log^2 n}{2^6 \cdot n}$, we find that $\mathcal{R}(\beta) < \frac{3}{8} \log^4(n/2)$. On the other hand, if $y_i = 1$, then $\mathcal{R}(\beta) > \frac{3}{4} \log^4 n/2$. Therefore, a 2-factor approximation to $\mathcal{R}(\beta)$ is able to decide if $y_i$ equals zero or one. By reduction to the INDEX problem, this implies that any 2-factor approximation to $\mathcal{R}(\beta)$ must take at least $\Omega(n)$ space (see Theorem 6).

Now we must just prove that $\mu_\mathbf{y}(\mathbf{X}) = \mathcal{O}(n)$. We will use the following inequality [13]. For any two length $n$ sequences of positive numbers, $a_r$, $r = 1 \ldots n$ and $b_r$, $r = 1 \ldots n$,

$$\frac{\sum_{r=1}^n a_r}{\sum_{r=1}^n b_r} \le \max_{r=1 \ldots n} \frac{a_r}{b_r},$$

where the maximum is taken over an arbitrary fixed ordering of the sequences. Let us define these two sequences as follows for a fixed $\beta \in \mathbb{R}^d$. For $i \in 1 \ldots n/2$, if $\mathbf{X}_i \beta > 0$, let $a_i = \mathbf{X}_i \beta$ and $b_i = -1 \cdot \mathbf{X}_{2i} \beta$. If $\mathbf{X}_i \beta < 0$, let $a_i = \mathbf{X}_{2i} \beta$ and $b_i = -1 \cdot \mathbf{X}_i \beta$. We can disregard the case where $\mathbf{X}_i \beta = 0$, since this will not affect the sums of the sequences. Given such sequences, we get:

$$\frac{\|(\mathbf{X}\beta)^+\|_1}{\|(\mathbf{X}\beta)^-\|_1} = \frac{\sum_r a_r}{\sum_r b_r} \le \max_r \frac{a_r}{b_r} \le \frac{1}{\mu}.$$

The last inequality follows since $\mathbf{X}_{2i} \beta = -\mu \cdot \mathbf{X}_i \beta$ for $i = 1 \ldots n/2$. Hence, we conclude that $\mu_{\mathbf{y}}(\mathbf{X}) = \mu^{-1} = \mathcal{O}(n)$. $\qquad \square$

The above theorem proves that a constant factor approximation to $\mathcal{R}(\cdot)$ requires $\Omega(\mu_{\mathbf{y}}(\mathbf{X}))$ space. We now extend this result to logistic loss.

**Theorem 3.** *Let $n \ge n_0$ (for some constant $n_0 \in \mathbb{N}$) be a positive integer. There exists a global constant $\epsilon_0 > 0$ and a matrix $\mathbf{X} \in \mathbb{R}^{n \times k}$ such that any data structure $\tilde{\mathcal{L}}(\cdot)$ that, with at least $2/3$ probability, satisfies (for a fixed $\beta \in \mathbb{R}^d$):*

$$(1 - \epsilon_0)\mathcal{L}(\beta; \mathbf{X}) \le \tilde{\mathcal{L}}(\beta; \mathbf{X}) \le (1 + \epsilon_0)\mathcal{L}(\beta; \mathbf{X})$$

*and requires at least $\tilde{\Omega}(d \cdot \mu_{\mathbf{y}}(\mathbf{X}))$ bits of space.*

*Proof.* The space complexity lower bound holds even if $\tilde{\mathcal{L}}(\cdot)$ approximates $\tilde{\mathcal{R}}(\cdot)$ only on the values of $\beta$ used to query the data structure in the proof of Lemma 2.4. Define this set as

$$\mathcal{B} = \{[\tilde{\mathbf{M}}_i, -4 \log^2 n]^T \in \mathbb{R}^{k'} \mid i = 1 \ldots n/2\},$$

where $\tilde{\mathbf{M}}$ is used to construct $\mathbf{X}$ in eqn. 5. Since $\mathcal{R}(\beta) \ge \mathbf{X}_i \beta$, we get

$$\mathcal{R}(\beta) \ge \|\tilde{\mathbf{M}}_i\|_2^2 - 4 \log^2 n \ge \frac{\log^4(n/2)}{4} - 4 \log^2 n.$$

Therefore, $\mathcal{R}(\beta) \ge 1$ for all $n \ge n_0$, where $n_0$ is some constant in $\mathbb{N}$. By Lemma 2.1, the space complexity of a data structure that achieves

$$(1 - \epsilon)\mathcal{L}(\beta) \le \tilde{\mathcal{L}}(\beta) \le (1 + \epsilon)\mathcal{L}(\beta)$$

for a fixed $\beta \in \mathcal{B}$ must be at least the space complexity of a data structure achieving

$$\mathcal{R}(\beta) \le \tilde{\mathcal{R}}(\beta) \le \frac{(1 + 3\epsilon)}{(1 - 3\epsilon)}\mathcal{R}(\beta)$$

for $\beta \in \mathcal{B}$. We can now solve for $\epsilon$ by setting $(1+3\epsilon)/(1-3\epsilon) = 2$. Therefore, from Lemma 2.4, we conclude that there exists a constant $\epsilon_0 > 0$ such that any data structure providing an $\epsilon_0$-relative approximation to the logistic loss requires at least $\tilde{\Omega}(n) = \tilde{\Omega}(\mu_{\mathbf{y}}(\mathbf{X}))$ space.

Finally, applying the argument used in Corollary 2 of constructing a matrix $\mathbf{Y}$, we achieve an $\tilde{\Omega}(d \cdot \mu_{\mathbf{y}}(\mathbf{X}))$ lower bound. $\qquad \square$

While prior work has show that mergeable coresets must include $\Omega(d \cdot \mu_{\mathbf{y}}(\mathbf{X}))$ points to attain a constant factor guarantee to logistic loss [17], our lower bound result holds for general data structures and applies even for data structure providing the weaker "for-each" guarantee, where the guarantee must hold for an arbitrary but fixed $\beta \in \mathbb{R}^d$ with a specified probability. The proof of the lower bound in [17] relies on constructing a matrix $\mathbf{A}$ that encodes $n$ bits such that the $i$-th bit can be recovered by adding some points to $\mathbf{A}$ and performing logistic regression on the new matrix. Hence, a mergeable coreset that compresses $\mathbf{A}$ can be used to solve the INDEX problem of size $n$. Meanwhile, our proof does not require constructing a regression problem but rather allows recovering the $i$-th bit by only observing an approximate value of the ReLU loss at a single fixed input vector for a fixed matrix $\mathbf{A}$. In addition to an arguably simpler proof, our approach needs weaker assumptions on the data structure. Therefore, our lower bound applies in more general settings, i.e., when sparsification is applied to $\mathbf{X}$.

# 3 Computing the complexity measure $\mu_{\mathbf{y}}(\mathbf{X})$ in polynomial time

We present an efficient algorithm to compute the data complexity measure $\mu_{\mathbf{y}}(\mathbf{X})$ of Definition 1 on real data sets, *refuting an earlier conjecture that this measure was hard to compute* [10]. The importance of this measure for logistic regression has been well-documented in prior work and further understanding its behavior on real-world data sets would help guide further improvements to coreset construction for logistic regression.

Prior work conjectured that $\mu_{\mathbf{y}}(\mathbf{X})$ was hard to compute and presented a polynomial time algorithm to approximate the measure to within a $\mathrm{poly}(d)$-factor (see Theorem 3 of Munteanu et al. [10]). We refute this conjecture by showing that the complexity measure $\mu_{\mathbf{y}}(\mathbf{X})$ can in fact be computed efficiently via linear programming, as shown in the following theorem. The specific LP formulation for computing a vector $\beta^*$ such that $\mu_{\mathbf{y}}(\mathbf{X}) = \frac{\|(\mathbf{D_y}\mathbf{X}\beta^*)^-\|_1}{\|(\mathbf{D_y}\mathbf{X}\beta^*)^+\|_1}$ is given in eqn. (9) in the Appendix.

**Theorem 4.** *If the complexity measure $\mu_{\mathbf{y}}(\mathbf{X})$ of Definition 1 is finite, it can be computed exactly by solving a linear program with $2n$ variables and $4n$ constraints.*

We conclude the section by noting that prior experimental evaluations of coreset constructions in [10, 7] relied on estimates of $\mu_{\mathbf{y}}(\mathbf{X})$ using the method provided by Munteanu et al. [10]. We will empirically compare how prior methods of estimating the complexity measure compare to our exact method in Section 4.

# 4 Experiments

We provide empirical evidence verifying the algorithm of Section 3 to exactly computes the classification complexity measure $\mu_{\mathbf{y}}(\mathbf{X})$ of Definition 1. We compare our results with the approximate sketching-based calculation of Munteanu et al. [10].

In order to estimate $\mu_{\mathbf{y}}(\mathbf{X})$ for a dataset using the sketching-based approach of Munteanu et al. [10], we create several smaller sketched datasets of a given full dataset of size $n \times d$ ($n$ rows and $d$ columns). We then use a modified linear program along the lines of Munteanu et al. [10]. These new datasets are created so that they have number of rows $n' = \mathcal{O}(d \log(d/\delta))$, for various values of $\delta \in [0, 1]$, so that with probability at least $1 - \delta$, the estimated $\mu_{\mathbf{y}}(\mathbf{X})$ will lie between some lower bound (given by $t$, the optimum value of the aforementioned linear program) and an upper bound (given by $t \cdot \mathcal{O}(d \log d)$). In order to solve the modified linear programs, we make use of the OR-tools[1].

**Synthetic data:** We create the synthetic dataset as follows. First, we construct the *full* data matrix $\mathbf{X} \in \mathbb{R}^{n \times d}$ by drawing $n = 10,000$ samples from the $d$-dimensional Gaussian distribution $\mathcal{N}(0, \mathbf{I}_d)$ with $d = 100$. We generate an arbitrary $\beta \in \mathbb{R}^d$ with $\|\beta\|_2 = 1$ and generate the posterior $p(\mathbf{y}_i|\mathbf{x}_i) = 1/(1 + \exp(-\beta^\top \mathbf{x}_i + \mathcal{N}(0, 1)))$. Finally, we create the labels $\mathbf{y}_i$ for all $i = 1 \ldots n$ by setting $\mathbf{y}_i$ to one if $p(\mathbf{y}_i|\mathbf{x}_i) > 0.5$ and to $-1$ otherwise.

Using the full data matrix $\mathbf{A} = \mathbf{D_y}\mathbf{X}$, we create several sketched data matrices having a number of rows equal to $n' = \mathcal{O}(d \log(d/\delta))$. We choose $\delta$ so that $n' \in \{512, 1024, 2048, 4096, 8192\}$. These values of $n'$ are chosen to be powers of two so that we can employ the Fast Cauchy Transform algorithm (`FastL1Basis` [1]) for sketching. The algorithm is meant to ensure that the produced sketch identifies $\ell_1$ well-conditioned bases for $\mathbf{A}$, which is a prerequisite for using the subsequent linear program to estimate $\mu_{\mathbf{y}}(\mathbf{X})$. (See Section C for details).

The results are presented in Figure 1a. For various values of $n'$, including when $n' = n = 10,000$, which we deem to be the original data size, we show the exact computation of $\mu_{\mathbf{y}}(\mathbf{X})$ on the sketched matrix using the linear program of Theorem 4. We also show the corresponding upper and lower bounds on $\mu_{\mathbf{y}}(\mathbf{X})$ of the full data set as estimated by the well-conditioned basis hunting approximation proposed by Munteanu et al. [10]. For the lower bound, we use the optimum value of the modified linear program as proposed in Munteanu et al. [10] and detailed in Section C. We set the upper bound by multiplying the lower bound by $d \log(d/\delta)$. Note that this upper bound is conservative, and the actual upper bound could be a constant factor higher, since the guarantees of Munteanu et al. [10] are only up to a constant factor. The presented results are an average over 20 runs of different sketches

---

[1]https://developers.google.com/optimization

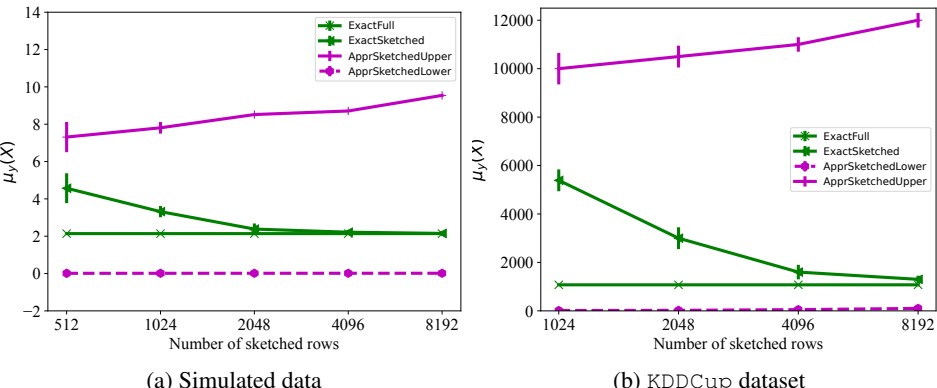

(a) Simulated data       (b) `KDDCup` dataset

Figure 1: Simulated data results for exact computation of $\mu_{\mathbf{y}}(\mathbf{X})$ (Theorem 4) using the full data (`Exactfull`), sketched data (`ExactSketched`) vs the approximate upper (`ApprSketchedUpper`) and lower bounds (`ApprSketchedLower`) as suggested by Munteanu et al. [10] (see Section C). The results clearly show that the upper and lower bounds can be very loose compared to our exact calculation of the complexity measure $\mu_{\mathbf{y}}(\mathbf{X})$

.

for each value of $n'$. Figure 1a shows that the exact computations on sketched matrices are close to the actual $\mu_{\mathbf{y}}(\mathbf{X})$ of the full data matrix, while the upper and lower bounds as proposed by Munteanu et al. [10] can be fairly loose.

**Real data experiments:** To test our setup on real data, we make use of the `sklearn KDDcup` dataset.[2] The dataset consists of 100654 data points and over 50 features. We only use continuous features which reduces the feature size to 33. The dataset contains 3377 positive data points, while the rest are negative. To create various sized subsets for exact calculation, we subsample from positives and negatives so that they are in about equal proportion. For larger subsamples, we retain all the positives, and subsample the rest from the negative data points. Since the calculation of $\mu$ for the full dataset is intractable as it will require to solve an optimization problem of over 400k constraints, we subsample 16384 data points and use its $\mu$ as proxy for that of the full dataset (referred to as `ExactFull`. For such a large subsample, the error bars are small. We compare against sketching and exact $\mu$ calculations for smaller subset (See Figure 1b for the results).

## 5 Future work

Our work shows that existing coresets are optimal up to lower order factors in the regime where $\mu_{\mathbf{y}}(\mathbf{X}) = \mathcal{O}(1)$. A clear open direction would be proving a space complexity lower bound that holds for all valid values of $\epsilon$ and $\mu_{\mathbf{y}}(\mathbf{X})$. Additionally, there is still a $d$ factor gap between existing upper bounds [9] and our lower bounds (Theorems 1 and 3) in the regime where $\epsilon$ is constant and the complexity measure varies or the complexity measure is constant and $\epsilon$ varies respectively.

Another interesting direction would be to explore whether additional techniques can further reduce the space complexity in approximating logistic loss compared to coresets alone. While Theorem 1 shows that the size of coreset constructions are essentially optimal, it does not preclude reducing the space by a $d$ factor by using other methods. In particular, the construction of $\mathbf{X}$ used in the proof of Theorem 1 is sparse, and so existing matrix sparsification methods would save this $d$ factor here.

We also note that our first lower bound (Theorem 1) only applies to data structures providing the the "for-all" guarantee on the logistic loss, i.e., with a given probability, the error guarantee in eqn.(2) holds for all $\beta \in \mathbb{R}^d$. It would be interesting to know if it could strengthened to apply in the "for-each" setting as Theorem 3 does, where $\beta \in \mathbb{R}^d$ is arbitrary but fixed.

Finally, it would be useful to gain a better understanding of the complexity measure $\mu_{\mathbf{y}}(\mathbf{X})$ in real data through more comprehensive experiments using our method provided in Theorem 4. In particular, subsampling points of a data set may create bias when computing $\mu_{\mathbf{y}}(\mathbf{X})$.

---

[2]`sklearn.datasets.fetch_kddcup99()` provides an API to access this dataset.

## Acknowledgments

We thank the anonymous reviewers for useful feedback on improving the presentation of this work. Petros Drineas and Gregory Dexter were partially supported by NSF AF 1814041, NSF FRG 1760353, and DOE-SC0022085. Rajiv Khanna was supported by the Central Indiana Corporate Partnership AnalytiXIN Initiative.

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

## A Preliminaries

In this section, we list some useful results from prior work.

### A.1 Hoeffding's inequality

We use the following formulation of Hoeffding's inequality.

**Theorem 5** (Theorem 2 in [4]). *Let $X_1, \ldots, X_n$ be independent random variables with $m_i \leq X_i \leq M_i$, $1 \leq i \leq n$. Then, for any $t > 0$,*

$$\mathbb{P}\left(\Big|\sum_{i=1}^{n}(X_i - \mathbb{E}[X_i])\Big| \geq t\right) \leq 2\exp\left(\frac{-2t^2}{\sum_{i=1}^{n}(M_i - m_i)^2}\right).$$

### A.2 `INDEX` problem

Both Theorem 1 and Theorem 3 rely on a reduction to the randomized `INDEX` problem. We define the `INDEX` problem as a data structure as done in [6].

**Definition 2.** *(`INDEX` problem) The `INDEX` data structure stores an input string $\mathbf{y} \in \{0,1\}^n$ and supports a query function, which receives input $i \in [n]$ and outputs $\mathbf{y}_i \in \{0,1\}$ which is the $i$-th bit of the underlying string.*

The following theorem provides a space complexity lower bound for the `INDEX` problem.

**Theorem 6.** *(see [8]) In the `INDEX` problem, suppose that the underlying string $\mathbf{y}$ is drawn uniformly from $\{0,1\}^n$ and the input $i$ of the query function is drawn uniformly from $[n]$. Any (randomized) data structure for `INDEX` that succeeds with probability at least $2/3$ requires $\Omega(n)$ bits of space, where the randomness is taken over both the randomness in the data structure and the randomness of $s$ and $i$.*

## B Proofs

**Proof for Lemma 2.1**

*Proof.* Let $\mathcal{R}_{\min} = \inf_{\beta \in \mathbb{R}^d} \mathcal{R}(\beta)$. We now prove that a data structure, $\tilde{\mathcal{L}}(\cdot)$, which approximates logistic loss to relative error can be used to construct an approximation to the ReLu loss, $\tilde{\mathcal{R}}(\cdot)$, by $\tilde{\mathcal{R}}(\beta) = \frac{1}{t} \cdot \tilde{\mathcal{L}}(t \cdot \beta)$ for large enough constant $t > 0$. To show this, we start by bounding the approximation error of the logistic loss for a single point. First, we derive the following inequality when $r > 0$:

$$\frac{1}{t}\log(1 + e^{t \cdot r}) - r = \frac{1}{t}\left(\log(e^{rt}) + \log\left(\frac{1 + e^{rt}}{e^{rt}}\right)\right) - r = \frac{1}{t} \cdot \log\left(1 + \frac{1}{e^{rt}}\right) \leq \frac{1}{t \cdot e^{rt}}.$$

Therefore, if $\mathbf{x}_i^T\beta > 0$, then $|\frac{1}{t}\log(1 + e^{t \cdot \mathbf{x}_i^T\beta}) - \mathbf{x}_i^T\beta| < \frac{1}{t \cdot e^{t \cdot \mathbf{x}_i^T\beta}}$. Next, we consider the case where $r \leq 0$, in which case $\text{ReLu}(r) = 0$. It directly follows that $0 \leq \frac{1}{t}\log(1 + e^{t \cdot r}) \leq \frac{e^{t \cdot r}}{t}$. Therefore,

$$|\frac{1}{t} \cdot \log(1 + e^{t \cdot r}) - \max\{0, r\}| \leq \frac{1}{t \cdot e^{t \cdot |r|}} \leq \frac{1}{t}.$$

We use these inequalities to bound the difference in the transformed logistic loss and ReLu loss as follows:

$$|\frac{1}{t} \cdot \mathcal{L}(t \cdot \beta) - \mathcal{R}(\beta)| = \Big|\sum_{i=1}^{n}\frac{1}{t}\log(1 + e^{t \cdot \mathbf{x}_i^T\beta}) - \max\{0, \mathbf{x}_i^T\beta\}\Big|$$

$$\leq \sum_{i=1}^{n}|\frac{1}{t}\log(1 + e^{t \cdot \mathbf{x}_i^T\beta}) - \max\{0, \mathbf{x}_i^T\beta\}| \leq \frac{n}{t}. \tag{6}$$

Therefore, if we set $t^* = \frac{n}{\epsilon \cdot \mathcal{R}_{\min}(\beta)}$, then

$$|\frac{1}{t^*}\mathcal{L}(t^* \cdot \beta) - \mathcal{R}(\beta)| \leq \epsilon \mathcal{R}_{\min}(\beta) \leq \epsilon \mathcal{R}(\beta), \tag{7}$$

for all $\beta \in \mathcal{B}$. We can then show that $\tilde{\mathcal{L}}(\cdot)$ can be used to approximate $\mathcal{R}(\cdot)$ by applying the triangle inequality, the error guarantee of $\tilde{\mathcal{L}}(\cdot)$ and eqn. 7. For all $\beta \in \mathcal{B}$:

$$\left|\frac{1}{t^*}\cdot\tilde{\mathcal{L}}(t^* \cdot \beta) - \mathcal{R}(\beta)\right| \leq \left|\frac{1}{t^*}\cdot\tilde{\mathcal{L}}(t^* \cdot \beta) - \frac{1}{t^*}\cdot\mathcal{L}(t^* \cdot \beta)\right| + \left|\frac{1}{t^*}\cdot\mathcal{L}(t^* \cdot \beta) - \mathcal{R}(\beta)\right|$$

$$\leq \epsilon \cdot \frac{1}{t^*}\cdot\mathcal{L}(t^* \cdot \beta) + \epsilon \mathcal{R}(\beta)$$

$$\leq \epsilon \cdot \left(\mathcal{R}(\beta) + \left|\mathcal{R}(\beta) - \frac{1}{t^*}\cdot\mathcal{L}(t^* \cdot \beta)\right|\right) + \epsilon \mathcal{R}(\beta)$$

$$\leq \epsilon \cdot \mathcal{R}(\beta) + \epsilon^2 \cdot \mathcal{R}(\beta) + \epsilon \cdot \mathcal{R}(\beta) \leq 3\epsilon \mathcal{R}(\beta).$$

Note that we can set $\mathcal{R}_{\min}$ to one and achieve the same error guarantee, since $\inf_{\beta \in \mathcal{B}} \mathcal{R}(\beta) > 1$. Therefore, storing $t^*$ takes $\mathcal{O}(\log \frac{n}{\epsilon})$ space.

$\square$

**Proof for Lemma 2.2**

*Proof.* We will first lower bound the space needed by any data structure which approximates ReLu loss to $\epsilon$-relative error. Later, we will show that this implies a lower bound on the space complexity of any data structure $f(\cdot)$ for approximating logistic loss. Let $\tilde{\mathcal{R}}(\cdot)$ approximate $\mathcal{R}(\cdot)$ such that $\mathcal{R}(\beta) \leq \tilde{\mathcal{R}}(\beta) \leq (1 + \epsilon)\mathcal{R}(\beta)$ for all $\beta \in \mathbb{R}^d$. We can rewrite $\mathcal{R}(\beta)$ as follows:

$$\mathcal{R}(\beta) = \sum_{i=1}^{n} \max\{0, \mathbf{x}_i^T\beta\} = \sum_{i=1}^{n} \nicefrac{1}{2} \cdot \mathbf{x}_i^T\beta + \nicefrac{1}{2} \cdot |\mathbf{x}_i^T\beta| = \frac{1}{2}\mathbf{1}^T\mathbf{X}\beta + \frac{1}{2}\|\mathbf{X}\beta\|_1.$$

We next use the fact that $\mathcal{R}(\beta) \leq \|\mathbf{X}\beta\|_1$ to get

$$|\mathcal{R}(\beta) - \tilde{\mathcal{R}}(\beta)| = |\frac{1}{2}\mathbf{1}^T\mathbf{X}\beta + \frac{1}{2}\|\mathbf{X}\beta\|_1 - \tilde{\mathcal{R}}(\beta)| \leq \epsilon \mathcal{R}(\beta)$$

$$\Rightarrow |\frac{1}{2}\mathbf{1}^T\mathbf{X}\beta + \frac{1}{2}\|\mathbf{X}\beta\|_1 - \tilde{\mathcal{R}}(\beta)| \leq \epsilon\|\mathbf{X}\beta\|_1.$$

We can store $\mathbf{1}^T\mathbf{X}$ exactly in $\mathcal{O}(d)$ space as a length $d$ vector. We define a new data structure $\mathcal{H}(\cdot)$ such that $\mathcal{H}(\beta) = 2\tilde{\mathcal{R}}(\beta) - \mathbf{1}^T\mathbf{X}\beta$, and, using the above inequality, $\mathcal{H}(\beta)$ satisfies:

$$|\|\mathbf{X}\beta\|_1 - \mathcal{H}(\beta)| \leq 2\epsilon\|\mathbf{X}\beta\|_1,$$

for all $\beta \in \mathbb{R}^d$. Therefore, $\mathcal{H}(\beta)$ is an $\epsilon$-relative approximation to $\|\mathbf{X}\beta\|_1$ after adjusting for constants and solves the $\ell_1$-subspace sketch problem (see Definition 1.1 in [6]). By Corollary 3.13 in [6], the data structure $\mathcal{H}(\cdot)$ requires $\tilde{\Omega}\left(\frac{d}{\epsilon^2}\right)$ bits of space if $d = \Omega(\log 1/\epsilon)$ and $n = \tilde{\Omega}\left(d\epsilon^{-2}\right)$. Therefore, we conclude that any data structure which approximates $\mathcal{R}(\beta)$ to $\epsilon$-relative error for all $\beta \in \mathbb{R}^d$ with at least $2/3$ probability must use $\tilde{\Omega}\left(\frac{d}{\epsilon^2}\right)$ bits in the worst case.

The proof in [6] that leads to Corollary 3.13 proceeds by constructing a matrix $\mathbf{A}$ ($\mathbf{X}$ in our notation) and showing that $\epsilon$-relative error approximations to $\|\mathbf{A}\beta\|_1$ require the stated space complexity. We next show that, $\mu_{\mathbf{y}}(\mathbf{A}) \leq 4$. The construction of $\mathbf{A}$ is described directly above Lemma 3.10. This matrix $\mathbf{A}$ is first set to contain all $2^k$ unique $k$ length vectors in $\{-1, 1\}^k$ for some value of $k$. Each row of $\mathbf{A}$ is then re-weighted: specifically, the $i$-th row of $\mathbf{A}$ is re-weighted by some $y_i \in [2\sqrt{k}, 8\sqrt{k}]$.

For any vector $\beta \in \mathbb{R}^d$, $\|(\mathbf{A}\beta)^+\|_1 \leq 4 \cdot \|(\mathbf{A}\beta)^-\|_1$ by the following argument. For an arbitrary $i \in [n]$, let $\mathbf{A}_i = y_i \cdot \mathbf{v}$ where $\{\pm 1\}^k$. Then there exists a unique $j \in [n]$ such that $\mathbf{A}_j = -1 \cdot y_j \cdot \mathbf{v}$, and so $\mathbf{A}_j\beta < 0$. Furthermore, since $y_j \geq 2\sqrt{k}$ and $y_i \leq 8\sqrt{k}$, $\frac{|\mathbf{A}_i\beta|}{|\mathbf{A}_j\beta|} \leq 4$. By applying this argument to each row of $\mathbf{A}$ where $\mathbf{A}_i\beta > 0$, we conclude that $\|(\mathbf{A}\beta)^+\|_1 \leq 4 \cdot \|(\mathbf{A}\beta)^-\|_1$, and hence $\mu_{\mathbf{y}}(\mathbf{X}) \leq 4$.

Finally, we show that $\mathcal{R}(\beta; \mathbf{X})$ is lower bounded in this construction. Given any vector $\beta \in \mathbb{R}^d$, there exists a row of $\mathbf{X}$ such that $\mathbf{X}_{ij} \cdot \beta_j \geq 0$ for all $j \in \{1 \dots d\}$, that is, the $j$-th entry of $\mathbf{X}_i$ has the same sign as $\beta_j$ in all non-zero entries of $\beta$. Therefore, $\mathbf{X}_i\beta = y_i \cdot \|\beta\|_1$. Since $y_i \geq 3\sqrt{k}$, we conclude that $\mathcal{R}(\beta; \mathbf{X}) \geq 3\|\beta\|_1$. $\qquad\square$

**Proof of Lemma 2.3**

*Proof.* Let $\mathbf{A} \in \{-1, 1\}^{n \times k}$ be a random matrix where each entry is uniformly sampled from $\{-1, 1\}$ in independent identical trials. Let $\{R_r\}_{r \in [k]}$ be independent Rademacher random variables. Then,

$$\mathbb{P}(|\langle \mathbf{A}_i, \mathbf{A}_j\rangle| \geq t) = \mathbb{P}\left(|\sum_{r=1}^{k} R_r| \geq t\right) \leq 2\exp\left(\frac{-t^2}{2k}\right),$$

where the last step follows from applying Hoeffding's inequality. There are fewer than $n^2$ tuples $(i, j) \in \{1 \dots d\} \times \{1 \dots d\}$ such that $i \neq j$. Therefore, by an application of the union bound,

$$\mathbb{P}(|\langle \mathbf{A}_i, \mathbf{A}_j\rangle| \geq t \text{ for all } i \neq j) \leq n^2 \cdot 2\exp\left(\frac{-t^2}{2k}\right).$$

Setting $t = 4\sqrt{k} \cdot \log n$, we see that the right side of the inequality is less than one whenever $n > 1$. Therefore, there must exist a matrix $\mathbf{M}$ satisfying the specified criteria. $\qquad\square$

**Proof of Theorem 4**

*Proof.* We now derive a linear programming formulation to compute the complexity measure in Definition 1. Note that we flip the numerator and denominator from Definition 1 without loss of generality. Let $\beta^* \in \mathbb{R}^d$ be:[3]

$$\beta^* = \operatorname*{argmax}_{\|\beta\|_2=1} \frac{\|(\mathbf{D_y}\mathbf{X}\beta)^-\|_1}{\|(\mathbf{D_y}\mathbf{X}\beta)^+\|_1}$$

$$\Rightarrow \mu_\mathbf{y}(\mathbf{X}) = \frac{\|(\mathbf{D_y}\mathbf{X}\beta^*)^-\|_1}{\|(\mathbf{D_y}\mathbf{X}\beta^*)^+\|_1}$$

The second line above uses the fact that the definition of $\mu_\mathbf{y}(\mathbf{X})$ does not depend on the scaling of $\beta$. If $C$ is an arbitrary positive constant, then there exists a constant $c > 0$ such that:

$$c \cdot \beta^* = \operatorname*{argmax}_{\beta \in \mathbb{R}^d} \|(\mathbf{D_y}\mathbf{X}\beta)^-\|_1 \text{ such that } \|(\mathbf{D_y}\mathbf{X}\beta)^+\|_1 \leq C$$

$$= \operatorname*{argmax}_{\beta \in \mathbb{R}^d} \|\mathbf{D_y}\mathbf{X}\beta\|_1 \text{ such that } \|(\mathbf{D_y}\mathbf{X}\beta)^+\|_1 \leq C.$$

Again, $\|(\mathbf{D_y}\mathbf{X}\beta^*)^-\|_1 / \|(\mathbf{D_y}\mathbf{X}\beta^*)^+\|_1$ is invariant to rescaling of $\beta^*$, so we may assume that $c$ is equal to one without loss of generality. We now reformulate the last constraint as follows:

$$\|(\mathbf{D_y}\mathbf{X}\beta)^+\|_1 = \sum_{i=1}^{n} \max\{[\mathbf{D_y}\mathbf{X}\beta]_i, 0\}$$

$$= \sum_{i=1}^{n} \frac{1}{2}[\mathbf{D_y}\mathbf{X}\beta]_i + \frac{1}{2}|[\mathbf{D_y}\mathbf{X}\beta]_i|$$

$$= \frac{1}{2}\mathbf{1}^T\mathbf{D_y}\mathbf{X}\beta + \frac{1}{2}\|\mathbf{D_y}\mathbf{X}\beta\|_1.$$

Therefore, the above formulation is equivalent to:

$$\beta^* = \operatorname*{argmax}_{\beta \in \mathbb{R}^d} \|\mathbf{D_y}\mathbf{X}\beta\|_1 \quad \text{such that } \frac{1}{2}\mathbf{1}^T\mathbf{D_y}\mathbf{X}\beta + \frac{1}{2}\|\mathbf{D_y}\mathbf{X}\beta\|_1 \leq C$$

$$= \operatorname*{argmin}_{\beta \in \mathbb{R}^d} \mathbf{1}^T\mathbf{D_y}\mathbf{X}\beta \text{ such that } \|\mathbf{D_y}\mathbf{X}\beta\|_1 \leq C.$$

---

[3]For notational simplicity, we assume $\beta^*$ is unique, but this is not necessary.

Next, we replace $\mathbf{D_y X}\beta$ with a single vector $\mathbf{z} \in \mathbb{R}^n$ and a linear constraint to guarantee that $\mathbf{z} \in \text{Range}(\mathbf{D_y X})$. Let $\mathbf{P}_R \in \mathbb{R}^{n \times n}$ be the orthogonal projection to $\text{Range}(\mathbf{D_y X})$. Then,

$$\mathbf{z}^* = \underset{\mathbf{z} \in \mathbb{R}^d}{\arg\min} \ \mathbf{1}^T \mathbf{z} \tag{8}$$
$$\text{such that } \|\mathbf{z}\|_1 \leq C \quad \text{and} \quad (\mathbf{I} - \mathbf{P}_R)\mathbf{z} = \mathbf{0}.$$

Next, we solve this formulation by constructing a linear program such that $[\mathbf{z}_+, \mathbf{z}_-] \in \mathbb{R}^{2n}$ corresponds to the absolute value of the positive and negative elements of $\mathbf{z}$, namely

$$\mathbf{z}^* = \underset{\beta \in \mathbb{R}^d}{\arg\min} \ \mathbf{1}_n^T(\mathbf{z}_+ - \mathbf{z}_-) \tag{9}$$
$$\text{such that } \mathbf{1}_n^T(\mathbf{z}_+ + \mathbf{z}_-) \leq C \quad \text{and} \quad (\mathbf{I} - \mathbf{P}_R)(\mathbf{z}_+ - \mathbf{z}_-) = \mathbf{0} \quad \text{and} \quad \mathbf{z}_+, \mathbf{z}_- \geq 0.$$

Observe that this is a linear program with $2n$ variables and $4n$ constraints. After solving this program for $\mathbf{z}_+^*$ and $\mathbf{z}_-^*$, we can compute $\mathbf{z}^* = \mathbf{z}_+^* - \mathbf{z}_-^*$. From this, we can compute $\beta^*$ by solving the linear system $\mathbf{z}^* = \mathbf{D_y X}\beta^*$, which is guaranteed to have a solution by the linear constraint $(\mathbf{I} - \mathbf{P}_R)\mathbf{z}^* = \mathbf{0}$.

After solving for $\beta^*$, we can compute

$$\mu_\mathbf{y}(\mathbf{X}) = \frac{\|(\mathbf{D_y X}\beta^*)^-\|_1}{\|(\mathbf{D_y X}\beta^*)^+\|_1},$$

thus completing the proof. $\qquad\square$

## C  Modified linear program

For completeness, we reproduce the linear program of Munteanu et al. [10][Section A] to estimate the complexity measure $\mu_\mathbf{y}(\mathbf{X})$:

$$\begin{aligned}
\min \ & \sum_{i=1}^n b_i \\
\text{s.t. } & \forall i \in [n] : (U\beta)_i = a_i - b_i \\
& \forall i \in [d] : \beta_i = c_i - d_i \\
& \sum_{i=1}^d c_i + d_i \geq 1 \\
& \forall i \in [n] : a_i, b_i \geq 0 \\
& \forall i \in [d] : c_i, d_i \geq 0.
\end{aligned}$$

Here, $U$ is $\ell_1$-well-conditioned basis [2] of the matrix $\mathbf{D}_y\mathbf{X}$. Because of the well-conditioned property, $\mu$ can be estimated to be within the bounds:

$$\frac{1}{t} \leq \mu_\mathbf{y}(\mathbf{X}) \leq \text{poly}(d).\frac{1}{t},$$

where $t = \min_{\|\beta\|_1=1} \|(U\beta)^-\|_1$, and recall that $d$ is the number of columns of $\mathbf{X}$. Munteanu et al. [10] designed the above linear program to solve for $t$. However, note that trivially the LP as it is written could be trivially solved with $\forall i c_i = d_i \implies \beta_i = 0 \implies a_i = b_i = 0$, which unfortunately gives $t = 0$ always trivially. To get around this problem, we modify the above program as follows:

$$\min \quad \sum_{i=1}^{n} b_i$$

$$\text{s.t. } \forall i \in [n] : (U\beta)_i = a_i - b_i$$
$$\forall i \in [d] : h_i = c_i - \beta_i$$
$$\forall i \in [d] : h_i = d_i + \beta_i$$
$$\forall i \in [d] : c_i \leq M v_i$$
$$\forall i \in [d] : d_i \leq M(1 - v_i)$$
$$\sum_{i=1}^{d} h_i \geq 1$$
$$\forall i \in [n] : a_i, b_i \geq 0$$
$$\forall i \in [d] : c_i, d_i, h_i \geq 0$$
$$\forall i \in [d] : v_i \in \{0, 1\}$$

Here, $M$ is a sufficiently large value as is often used for Big-M constraints[4] in linear programs. The variable $h_i$ simulates the $|\beta_i|$, and is set according to the binary variable $v_i$ which decides which one of $c_i$ or $d_i$ is 0. Further, note that $\sum_i h_i \geq 1$ is equivalent to $\sum_i h_i = 1$ here since scaling down the norm of $\beta$ can only bring the optimization cost down. To solve the optimization problem, we use the `Google OR-tools` and the wrapper `pywraplp` with the solver `SAT` which can handle integer programs since the above program is no longer a pure linear program because of the binary variables $v_i$.

## D  Low rank approximation to logistic loss

Here, we provide a very simple data structure that provides an additive error approximation to the logistic loss. While the method is straightforward, we are unaware of this approximation being specified in prior work, and it may be useful in the natural setting where the input matrix $\mathbf{X}$ has low stable rank.

We show that any low-rank approximation $\bar{\mathbf{X}}$ of the data matrix $\mathbf{X}$ can be used to approximate the logistic loss function $\mathcal{L}(\beta)$ up to a $\sqrt{n}\|\mathbf{X} - \bar{\mathbf{X}}\|_2 \|\beta\|_2$ additive error. The factor $\|\mathbf{X} - \bar{\mathbf{X}}\|_2 \|\beta\|_2$ is the spectral norm (or two-norm) error of the low-rank approximation and we also prove that this bound is tight in the worst case. Low rank approximations are commonly used to reduce the time and space complexity of numerical algorithms, especially in settings where the data matrix $\mathbf{X}$ is numerically low-rank or has a decaying spectrum of singular values.

Using low-rank approximations of $\mathbf{X}$ to estimate the logistic loss is appealing due to the extensive body work on fast constructions of low-rank approximations via sketching, sampling, and direct methods [5]. We show that a spectral approximation provides an additive error guarantee for the logistic loss and that this guarantee is tight on worst-case inputs.

**Theorem 7.** *If* $\mathbf{X}, \tilde{\mathbf{X}} \in \mathbb{R}^{n \times d}$, *then for all* $\beta \in \mathbb{R}^d$,

$$|\mathcal{L}(\beta; \mathbf{X}) - \mathcal{L}(\beta, \tilde{\mathbf{X}})| \leq \sqrt{n}\|\mathbf{X} - \tilde{\mathbf{X}}\|_2 \|\beta\|_2.$$

---

[4]See  Linear and Nonlinear Optimization (2nd ed.).  Society for Industrial Mathematics

*Proof.* To simplify the notation, let $\mathbf{s} = \mathbf{X}\beta$ and $\mathbf{d} = (\mathbf{X} - \tilde{\mathbf{X}})\beta$. We can then write the difference in the log loss as:

$$|\mathcal{L}(\beta; \mathbf{X}) - \mathcal{L}(\beta, \tilde{\mathbf{X}})| = \left( \sum_{i=1}^{n} \log\left(1 + e^{\mathbf{x}_i^T \beta}\right) + \frac{\lambda}{2}\|\beta\|_2^2 \right) - \left( \sum_{i=1}^{n} \log\left(1 + e^{\tilde{\mathbf{x}}_i^T \beta}\right) + \frac{\lambda}{2}\|\beta\|_2^2 \right)$$

$$= \left| \sum_{i=1}^{n} \log\left( \frac{1 + e^{\mathbf{s}_i}}{1 + e^{\mathbf{s}_i + \mathbf{d}_i}} \right) \right|$$

$$\leq \left| \sum_{i=1}^{n} \log\left( \frac{1 + e^{\mathbf{s}_i}}{1 + e^{\mathbf{s}_i - |\mathbf{d}_i|}} \right) \right|$$

$$= \left| \sum_{i=1}^{n} \log\left( \frac{1 + e^{\mathbf{s}_i}}{1 + e^{-|\mathbf{d}_i|}e^{\mathbf{s}_i}} \right) \right|$$

$$\leq \left| \sum_{i=1}^{n} \log\left( \frac{1}{e^{-|\mathbf{d}_i|}} \frac{1 + e^{\mathbf{s}_i}}{1 + e^{\mathbf{s}_i}} \right) \right|$$

$$= \sum_{i=1}^{n} |\mathbf{d}_i| = \|\mathbf{d}\|_1.$$

Therefore, we can conclude that $|\mathcal{L}(\beta; \mathbf{X}) - \mathcal{L}(\beta; \tilde{\mathbf{X}})| \leq \|\mathbf{d}\|_1 \leq \sqrt{n}\|\mathbf{d}\|_2 \leq \sqrt{n}\|\tilde{\mathbf{X}} - \mathbf{X}\|_2\|\beta\|_2$. $\square$

We note that Theorem 7 holds for any matrix $\tilde{\mathbf{X}} \in \mathbb{R}^{n \times d}$ that approximates $\mathbf{X}$ with respect to the spectral norm, and does not necessitate that $\tilde{\mathbf{X}}$ has low-rank. We now provide a matching lower-bound for the logistic loss function in the same setting.

**Theorem 8.** *For every $d, n \in \mathbb{N}$ where $d \geq n$, there exists a data matrix $\mathbf{X} \in \mathbb{R}^{n \times d}$, label vector $\mathbf{y} \in \{-1, 1\}^n$, parameter vector $\beta \in \mathbb{R}^d$, and spectral approximation $\tilde{\mathbf{X}} \in \mathbb{R}^{n \times d}$ such that:*

$$|\mathcal{L}(\beta; \bar{\mathbf{X}}) - \mathcal{L}(\beta; \mathbf{X})| \geq (1 - \delta)\sqrt{n}\|\mathbf{X} - \bar{\mathbf{X}}\|_2\|\beta\|_2,$$

*for every $\delta > 0$. Hence, the guarantee of Theorem 7 is tight in the worst case.*

*Proof.* To prove the theorems statement, we first consider the case of square matrices ($d = n$). In particular, first consider the case where $d = n = 1$, where $\mathbf{X} = [x]$ and $\tilde{\mathbf{X}} = [x + s]$, in which case $\|\mathbf{X} - \tilde{\mathbf{X}}\|_2 = s$. Then,

$$\lim_{x \to \infty} \mathcal{L}([1]; [x + s]) - \mathcal{L}([1]; [x]) = \lim_{x \to \infty} \log(1 + e^{x+s}) - \log(1 + e^x) = s$$

Which shows that for $\beta = [1]$ and $x$ with large enough magnitude $\mathcal{L}(\beta; \tilde{\mathbf{X}}) - \mathcal{L}(\beta; \mathbf{X}) = (1 - \delta)\|\mathbf{X} - \tilde{\mathbf{X}}\|_2$. Next, let $\mathbf{X} = x \cdot \mathbf{I}_n$, $\tilde{\mathbf{X}} = (x + s) \cdot \mathbf{I}_n$, and $\beta = \mathbf{1}_n$. Then for all $i \in [n]$, $\mathbf{x}_i^T \beta = x$ and $\mathbf{x}_i^T \beta = x + s$. Therefore,

$$\lim_{x \to \infty} \mathcal{L}(\beta; \tilde{\mathbf{X}}) - \mathcal{L}(\beta; \mathbf{X}) = \lim_{x \to \infty} \sum_{i=1}^{n} \left[ \log(1 + e^{x+s}) - \log(1 + e^x) \right] = sn.$$

Since $\|\mathbf{X} - \tilde{\mathbf{X}}\|_2 = \|s \cdot \mathbf{I}\|_2 = s$. $\|\beta\|_2 = \sqrt{n}$,

$$\lim_{x \to \infty} \mathcal{L}(\beta; \tilde{\mathbf{X}}) - \mathcal{L}(\beta; \mathbf{X}) = sn = \sqrt{n}\|\mathbf{X} - \tilde{\mathbf{X}}\|_2\|\beta\|_2$$

Hence, we conclude the statement of the theorem for the case where $d = n$. To conclude the case for $d \geq n$, note that $\sqrt{n}\|\mathbf{X} - \bar{\mathbf{X}}\|_2\|\beta\|_2$ does not change if we extend $\mathbf{X}$ and $\bar{\mathbf{X}}$ with columns of zeroes and extend $\beta$ with entries of zero until $\mathbf{X}, \bar{\mathbf{X}} \in \mathbb{R}^{n \times d}$ and $\mathbb{R}^d$. This procedure also does not change the loss at $\beta$, hence we conclude the statement of the theorem. $\square$

