# OpenReview forum: "The Space Complexity of Approximating Logistic Loss"
_NeurIPS.cc/2024/Conference — NeurIPS 2024 poster_

### Official Review · Reviewer_iEdU · 2024-07-07

**Soundness:** 4
**Presentation:** 3
**Contribution:** 4
**Rating:** 7
**Confidence:** 3

**Summary:**

This paper gives a d \eps^{-2} bounds for coresets of points/labels that preserve logistical functions. Such a bound is similar to the bounds obtained in some recent upper bounds, and the paper experimentally verifies that a natural complexity measure is likely needed for constructing such coresets. The paper also gives an algorithm for estimating this complexity measure.

**Strengths:**

Logistical functions are widely used in a multitude of learning tasks, and coresets have been studied for a variety of numerical/optimization problems.

The bounds shown are likely the right ones, and significantly demystifies the complexity measure.

**Weaknesses:**

The results doesn't cover the `for this solution' sparsification that's weaker than the 'for all' sparsfication, but is still sufficient in many applications. However, that version will likely require significantly different approaches.

**Questions:**

Are there wider classes of functions that the approach for showing coreset lower bounds here can generalize to?

**Limitations:**

yes

---

> ### Author Rebuttal · Authors · 2024-08-06
>
> > The results doesn't cover the `for this solution' sparsification that's weaker than the 'for all' sparsfication, but is still sufficient in many applications. However, that version will likely require significantly different approaches
>
> This is a great point, and we agree with the reviewer: it is not obvious how to prove the proposed quantification. This will probably require new approaches. In the final version, we will mention this as an open problem for future work.
>
> > Are there wider classes of functions that the approach for showing coreset lower bounds here can generalize to?
>
> Yes, our bounds also apply to approximations of ReLU loss, as our proof operates by first showing that an $\epsilon$-relative error approximation to logistic loss provides an $O(\epsilon)$-relative error approximation to ReLU loss. It seems likely that our lower bound would apply to the class of “hinge-like loss functions” defined by Mai et al (Definition 7 in their paper).

---

> > ### Comment · Reviewer_iEdU · 2024-08-12
> > **thank you**
> >
> > Thank you for these clarifications and further details. My opinions on the paper are unchanged.

---

### Official Review · Reviewer_9S9F · 2024-07-12

**Soundness:** 3
**Presentation:** 3
**Contribution:** 3
**Rating:** 6
**Confidence:** 4

**Summary:**

The paper gives mainly lower bounds against any data compression for logistic regression parameterized by a complexity measure $\mu$. They prove lower bounds $\Omega(d/\epsilon^2)$ as well as $\Omega(d\mu)$. Both are derived by reduction from the indexing indexing problem and a direct sum argument to extend the bounds by the d factor.

There are also new algorithms for computing $\mu$ by a linear program, and additive error bounds for low rank approximation for logistic regression. I found the latter more interesting than the former but it is completely hidden in the appendix.

**Strengths:**

* nearly tight lower bounds against data compression for logistic regression
* hold against arbitrary data reduction techniques, not only coresets
* well written paper

**Weaknesses:**

* state of the art discussion on upper bounds is slightly outdated: updating to the $O(d\mu/\epsilon^2)$ upper bounds https://arxiv.org/abs/2406.00328 will only strengten the contributions.
* the $\Omega(\mu)$ bound is only from a $\Omega(n)$ bound for some case where $\mu \approx n$

**Questions:**

* any chance the two bounds can be combined to a $\Omega(d\mu/\epsilon^2)$ bound?
* if not then do you think $O(d(\mu+1/\epsilon^2))$ might be possible?
* any chance the $\Omega(\mu)$ bound can be generalized to hold for arbitrary (or some range of) $\mu$, to give for instance $\Omega(\sqrt{n})$ when $\mu = \sqrt{n}$?
* to apply the direct sum argument over the $d$ indexing problems on the block diagonal matrix, is it not needed to apply some sort of union bound over the failure probability of each of the d indexing instances? If yes, is this possible in your reduction while losing only a $\log(d)$ factor or will the resulting bound only hold against 1/d failure probability?

**Limitations:**

see above

---

> ### Author Rebuttal · Authors · 2024-08-06
>
> > state of the art discussion on upper bounds is slightly outdated: updating to the $O(d\mu/\epsilon^2)$ upper bounds https://arxiv.org/abs/2406.00328 will only strengten the contributions.
>
> Thank you for pointing out this new result. This Arxiv submission was made after NeurIPS deadline, so we were not aware of this paper. We will indeed update our paper to compare to this tighter upper bound. Indeed, this strengthens the contributions of our work and we appreciate your comment and suggestion.
>
> > the $\Omega(n)$ bound is only from a $\Omega(n)$ bound for some case where $\mu \approx n$
>
> Our proof of $\Omega(\mu)$ space complexity using $\mu \approx n$ immediately extends to all values of $\mu \lesssim n$  by the fact we can always pad $\mathbf{X}$ with zeroes.
>
>
> > any chance the two bounds can be combined to a $\Omega(d\mu/\epsilon^2)$ bound?
>
> We believe that our two lower bounds will be informative for achieving an $\Omega(d\mu/\epsilon^2)$ bound. However, we did not find a way to readily bridge this gap. Both lower bounds work by reduction to the $\texttt{INDEX}$ problem, but in quite different ways. Intuitively, the difficulty is that the non-linearity of ReLU is used in the two proofs in different ways. Unfortunately, constructing a multiplicative trade-off between the two constructions is not straightforward.
>
>
> > if not then do you think $\Omega(d(\mu + \frac{1}{\epsilon^2}))$ might be possible?
>
> Yes, this lower bound can be derived from our two provided lower bounds by the fact that any approximation to the logistic loss on a matrix $\mathbf{C} = [\mathbf{A}, \mathbf{0}; \mathbf{0}, \mathbf{B}]$ must also provide an equally accurate approximation to the logistic loss of any query vector on either matrix $\mathbf{A}$ or $\mathbf{B}$. Hence, we obtain the lower bound you have mentioned by letting $\mathbf{A}$ be constructed from Theorem 1 and $\mathbf{B}$ be constructed from Theorem 3.
>
>
> > Any chance the $\Omega(n)$ bound can be generalized to hold for arbitrary (or some range of) $\mu$, to give for instance $\Omega(\sqrt{n})$ when $\mu \approx \sqrt{n}$?
>
> By using our construction and padding $\mathbf{X}$ with zeroes as described above, the lower bound can be immediately extended to all $\mu \lesssim n$.
>
>
> > to apply the direct sum argument over the $d$ indexing problems on the block diagonal matrix, is it not needed to apply some sort of union bound over the failure probability of each of the $d$ indexing instances? If yes, is this possible in your reduction while losing only a $\log(d)$ factor or will the resulting bound only hold against $1/d$ failure probability?
>
> Since we are going for a lower bound, the probabilistic nature of the guarantees actually works in our favor. A relative error approximation to the logistic loss on the diagonal block matrix $\mathbf{Y}$ simultaneously provides the relative error guarantee to each of its blocks $\mathbf{X}\_{(1)}...\mathbf{X}\_{(d)}$. Therefore, obtaining a relative error approximation for $\mathbf{Y}$ with probability $2/3$ requires that each $\mathbf{X}\_{(i)}$ is approximated with probability greater than $2/3$ ($2/3 = p^d$, where $p$ is the success probability of estimating $\mathbf{X}\_{(i)}$ to be precise).

---

> > ### Comment · Reviewer_9S9F · 2024-08-12
> >
> > Thank you for the rebuttal, I like the paper and will keep my score, which is already on the positive side.
> >
> > Two more comments on your rebuttal:
> > * I think Woodruff, Yasuda, 2023 had a more 'natural' way of providing LB for different values of $\mu$ rather than padding with zeros which you might want to consider
> > * your comment regarding $\Omega(d(\mu + \frac{1}{\epsilon^2}))$ is clearly straightforward from the two bounds. Actually, I was asking whether you believe in an $O(d(\mu + \frac{1}{\epsilon^2}))$ **upper** bound (in case there are principled difficulties for achieving $\Omega(d\mu/\epsilon^2)$)

---

> > > ### Author Response · Authors · 2024-08-13
> > >
> > > > I think Woodruff, Yasuda, 2023 had a more 'natural' way of providing LB for different values of $\mu$ rather than padding with zeros which you might want to consider
> > >
> > > Regarding Woodruff/Yasuda 2023: Is the reviewer referring to the extension of the $\mu = n$ case to $\mu < n$ for the $\Omega(\mu)$ bound using Woodruff/Yasuda 2023 rather than our padding technique? We agree that this is indeed worth exploring. However, it is not obvious to us (at least within this short time window) whether this is possible. We will keep thinking about this and we very much appreciate the reviewer's suggestion. Just as a final note, this would not alter the final bounds and our method is also valid and correct.
> > >
> > > > your comment regarding $\Omega(d(\mu + \frac{1}{\epsilon^2}))$ is clearly straightforward from the two bounds. Actually, I was asking whether you believe in an $O(d(\mu + \frac{1}{\epsilon^2}))$ upper bound (in case there are principled difficulties for achieving $\Omega(d\mu/\epsilon^2)$)
> > >
> > > From our perspective, it still seems feasible that an $\Omega(d\mu/\epsilon^2)$ lower bound holds and the difficulty stems from technical complexity in combining the two reductions smoothly.
> > >
> > > One intuitive insight related to our proofs is that the parameter $\mu$ describes how much better an $\epsilon$-relative error approximation to ReLU loss (which can be approximated by logistic loss) is than an $\epsilon$-relative error approximation to an $\ell\_1$-norm on worst case vectors in absolute terms. This is because $||(\mathbf{X}\beta)^+||\_1$ can be approximately $1/\mu$ of $||\mathbf{X}\beta||\_1$. However, this relation cannot hold for all vectors $\beta \in \mathbb{R}^d$ simultaneously. This cannot be immediately combined with the L1-norm lower bound, but may be useful to note in pursuing either an $O(d(\mu + 1/\epsilon^2))$ upper bound or $\Omega(d\mu/\epsilon^2)$ lower bound.

---

> > > > ### Comment · Reviewer_9S9F · 2024-08-13
> > > >
> > > > Thank you, I appreciate the additional comments.

---

### Official Review · Reviewer_hULF · 2024-07-12

**Soundness:** 3
**Presentation:** 2
**Contribution:** 3
**Rating:** 5
**Confidence:** 1

**Summary:**

The paper explores the space complexity lower bounds for data structures that approximate logistic loss with $\epsilon$-relative error in logistic regression problems. The authors provide an $\Omega(d/\epsilon^2)$ space complexity lower bound for datasets with constant $\mu$-complexity, demonstrating that existing coreset constructions are optimal within this regime up to lower order factors. They also establish a general $\Omega(d \cdot \mu_y(X))$ space lower bound for constant $\epsilon$, highlighting that the dependency on $\mu_y(X)$ is intrinsic and not an artifact of mergeable coresets. Additionally, the paper refutes a prior conjecture regarding the difficulty of computing $\mu_y(X)$ by presenting an efficient linear programming formulation. Empirical comparisons show that the proposed algorithm for computing $\mu_y(X)$ is more accurate than prior approximate methods. These contributions enhance the understanding of the fundamental limits and efficiencies in approximating logistic loss, providing insights that can guide future improvements in coreset constructions and related data structures.

**Strengths:**

The paper provides an $\tilde{\Omega}(d/\epsilon^2)$ space complexity lower bound when $\mu_y(x) = O(1)$ and a general $\tilde{\Omega}(d\cdot \mu_y(x))$ assuming $\epsilon$ is constant. Also, the paper refute a prior conjecture by providing a linear programming to compute $\mu_y(x)$.

**Weaknesses:**

N/A. The reviewer is not familiar with space complexity area.

**Questions:**

A minor issue, the font of the paper seems different from other papers.

**Limitations:**

Yes.

---

> ### Author Rebuttal · Authors · 2024-08-06
>
> > A minor issue, the font of the paper seems different from other papers.
>
> Thanks for pointing this out. We have fixed the issue.

---

> > ### Comment · Reviewer_hULF · 2024-08-10
> >
> > Thanks, I will keep my scores.

---

### Official Review · Reviewer_8cM9 · 2024-07-16

**Soundness:** 3
**Presentation:** 2
**Contribution:** 3
**Rating:** 5
**Confidence:** 2

**Summary:**

This paper establishes lower bounds on the space complexity for data structures approximating logistic loss with relative error, showing that existing methods are optimal up to lower order factors. It proves that any coreset providing an $(\varepsilon)$-relative error approximation to logistic loss requires storing $(\Omega(\frac{d}{\epsilon^2}))$ rows of the original data matrix. The paper also demonstrates that the dependency on the complexity measure $(\mu_y(X))$ is fundamental to the space complexity of approximating logistic loss. Further an efficient linear programming formulation is provided to compute $(\mu_y(X))$, refuting a prior conjecture about its computational difficulty.

**Strengths:**

- The paper introduces new space complexity lower bounds for data structures approximating logistic loss, which is a novel contribution in logistic regression. It also refutes a prior conjecture about the complexity measure \mu_y(X) and provides an efficient linear programming formulation.

- The paper is built upon existing work in the field. It provides rigorous theoretical proofs and empirical comparisons to prior methods, demonstrating the validity and effectiveness of the proposed approaches.

- The paper is clearly written, with a structured presentation of the problem, contributions, and results. Definitions and notations are provided to aid understanding, and the inclusion of related work helps contextualize the contributions.

**Weaknesses:**

- The paper should clearly articulate how its contributions differ from existing works, especially those by Munteanu et al. and Mai et al. Highlighting specific improvements or unique approaches.

- The experiments could have been conducted on more diverse datasets, including real-world applications beyond synthetic and KDDCup datasets.

- The paper introduces a new method to compute the complexity measure µy(X). Including more detailed explanations would be useful.

**Questions:**

-

**Limitations:**

Yes. Authors mention that more comprehensive experiments in computing $\mu_y(X)$ for real datasets as an open direction.

---

> ### Author Rebuttal · Authors · 2024-08-06
>
> > The paper should clearly articulate how its contributions differ from existing works, especially those by Munteanu et al. and Mai et al. Highlighting specific improvements or unique approaches.
>
> Our work is complementary to the works of Munteanu et al. and Mai et al. Both of those papers primarily consider providing coreset constructions for logistic loss with _upper_ bounds on the number of sampled points for relative error guarantees. On the other hand, the primary contributions of our work are providing _space complexity lower bounds_ for approximating logistic loss as well as an exact and tractable method (as opposed to being  NP Hard to evaluate, which was previously conjectured) to compute the complexity parameter $\mu$ that is present in both the upper bound of prior work and our lower bounds.
>
>
> The reviewer is correct that Munteanu et al. does have an $\Omega(n)$ space complexity lower bound when $\mu$ is unbounded, but that lower bound is superseded by the more recent results of Woodruff and Yasuda. The latter paper provides a lower bound for coreset construction. We compare our work to their result and explain how our result is more general in the contributions section, as well as in lines 263-266. In the final version, we will also add additional explanation after lines 263-266 on how the reduction used in the lower bound of Woodruff and Yasuda differs from ours.
>
> > The experiments could have been conducted on more diverse datasets, including real-world applications beyond synthetic and KDDCup datasets.
>
> We agree that further experiments on understanding the behavior of $\mu$ in real datasets would be an interesting problem, and we note this direction in our future works section. However, our work is focused on theoretically understanding the space complexity of approximating logistic loss. A thorough experimental exploration of the behavior of $\mu$ in real datasets would be an implementation-focused question and is not the main focus of this paper.
>
> > The paper introduces a new method to compute the complexity measure µy(X). Including more detailed explanations would be useful.
>
> We will add a section in the appendix of the final version with more explicit details on computing $\mu$ using the LP introduced in the proof of Theorem 4.

---

> > ### Comment · Reviewer_8cM9 · 2024-08-13
> >
> > I thank the authors for their response. I have updated my scores by 1.

---

### Comment · Area_Chair_yjk2 · 2024-08-09

Dear reviewers,

Thank you for taking the time to review the paper. Notice that the discussion period has begun, and will last until August 13 (4 more days).

During this time your active participation and engagement with the authors is very important, and highly appreciated. Specifically, please _read the responses_, _respond to them early on in the discussion_, and **discuss points of disagreement**.

Thank you for your continued contributions to NeurIPS 2024.

Best, Your Area Chair.

---

### Decision · Program_Chairs · 2024-09-25

**Decision:**

Accept (poster)

**Comment:**

This paper establishes lower bounds on the space complexity for data structures approximating logistic loss with relative error, showing that existing methods are optimal up to lower order factors.

The reviewers acknowledge that this paper introduces new space complexity lower bounds for data structures approximating logistic loss, which is a novel contribution in logistic regression. It also refutes a prior conjecture about the complexity measure $\mu_y(X)$ and provides an efficient linear programming formulation. Additionally, the reviewers commend the paper for its clarity and quality of writing.

Overall, this paper makes novel contributions. Therefore, I recommend accepting it, with the suggestion that the authors address the new literature pointed out by Reviewer 9S9F in the final version.